# Displaced Drude peak and bad metal from the interaction with slow fluctuations.

S. Fratini[1*], S. Ciuchi[2,3],

**1** Université Grenoble Alpes, CNRS, Grenoble INP, Institut Néel, 38000 Grenoble, France
**2** Dipartimento di Scienze Fisiche e Chimiche, Università dell'Aquila, 67100 Coppito (AQ), Italy
**3** Istituto dei Sistemi Complessi, CNR, 00185 Roma, Italy
* simone.fratini@neel.cnrs.fr

July 26, 2021

## Abstract

**Scattering by slowly fluctuating degrees of freedom can cause a transient localization of the current-carrying electrons in metals, driving the system away from normal metallic behavior. We illustrate and characterize this general phenomenon by studying how signatures of localization emerge in the optical conductivity of electrons interacting with slow bosonic fluctuations. The buildup of quantum localization corrections manifests itself in the emergence of a displaced Drude peak (DDP), whose existence strongly alters the low frequency optical response and suppresses the d.c. conductivity. We find that for sufficiently strong interactions, many-body renormalization of the fluctuating field induced at metallic densities enhances electron localization and the ensuing DDP phenomenon in comparison with the well-studied low concentration limit. Our results are compatible with the frequent observation of DDPs in electronic systems where slowly fluctuating degrees of freedom couple significantly to the charge carriers.**

## 1  Introduction

In many complex materials, generally termed *bad metals*, the electrical resistivity increases with temperature beyond the maximum values admissible by the semi-classical theory of transport [1–3]. Examples of bad metals include several classes of transition-metal oxides, including the cuprate superconductors and heavy fermion materials, as well as low-dimensional organic conductors [1, 4]. From a broader perspective, bad metal behavior can be viewed as a manifestation of a more complex phenomenon, where the anomalous behavior is not limited to charge transport alone: large values of the resistivity (i.e. low values of the conductivity at $\omega = 0$) are often accompanied by marked anomalies in the dynamical response of the material at frequencies $\omega > 0$, typically in the infra-red range [2, 3].

There is a broad class of bad metals where the Drude peak in the optical absorption — a key identifying feature of normal metals — is replaced by a prominent absorption peak located at finite frequency, signaling a marked breakdown of the conventional picture. While such anomalous peaks have now been observed for decades, theoretical attempts to explain their origin on a global scale have appeared only recently [5, 6], and our present understanding of the phenomenon is still limited. Many among the metallic compounds featuring displaced Drude peaks (DDP) are strongly correlated materials, which hints at a significant role played by electronic interactions in favoring the DDP phenomenology. Conversely, many correlated systems exist where the DDP is not observed. In fact, accurate theoretical results [7–9] highlighting bad metal behavior in the framework of the Hubbard

model — the paradigmatic model for correlated systems — have found no evidence of DDPs, implying that short range electronic correlations alone cannot explain the phenomenon, and additional ingredients must be at work. On even more fundamental grounds, it is unclear whether such displaced Drude peak embodies the resilient response of quasiparticles whose properties have been deeply altered by interactions [10], or if it originates instead from the optical absorption of emergent excitations unrelated to the original charge carriers [6, 11].

Here we analyze a general scenario that rationalizes the DDP observed in bad metals as the signature of quantum localization processes caused by a dynamic random environment: localization of the charge carriers suppresses their optical response at low frequencies, shifting the quasiparticle absorption to finite frequencies [12–15], as sketched in Fig. 1(a). This idea contrasts with the alternative view of DDPs originating from a separate absorption channel that is distinct from the Drude response of electronic carriers, as in the collective mode scenarios considered in Refs. [6, 11], Fig. 1(b). Furthermore, we propose that the randomness at the origin of the DDP is self-generated, i.e. not related to extrinsic sources of disorder, but rather caused by the existence of low-energy degrees of freedom that couple significantly to the charge carriers. These can be any of the various soft excitations that are ubiquitous in complex materials, such as lattice vibrations, magnetic fluctuations, collective charge excitations and critical modes near ordering transitions. Crucially, the dynamic nature of this type of disorder implies that localization processes are transient (i.e., limited in time), hampering but not precluding carrier diffusion: the d.c. conductivity does not vanish completely, yet it is strongly suppressed, favoring bad metal behavior, cf. Fig. 1(a).

The *transient localization* scenario described above has been thoroughly investigated in the last decade to address bad conduction in organic semiconductors [5, 16, 17], and more recently in halide perovskites [18], both featuring abundant low-frequency molecular vibrations related to their soft mechanical properties. This, together with the known fact that localization effects are maximum near band edges, i.e. precisely for those electronic states that are relevant in semiconductors, explains why the phenomenon is widespread in these systems. Here we demonstrate that the coupling to slow degrees of freedom is able to induce localization also in metals, where the charge conduction involves instead band states that are in principle much less sensitive to disorder. We find that the resulting DDP phenomenon is potentially more robust in metals than in semiconductors: this occurs as the intrinsic randomness related to the fluctuating environment is significantly enhanced by the back-reaction of the (large) electronic density, identifying a general physical mechanism that drives anomalous electronic properties in metals [19, 20].

## 2 Model and methods

To keep the discussion as general as possible we consider the simplest model describing lattice electrons interacting with bosonic degrees of freedom with a characteristic frequency scale $\omega_0$, i.e. the Holstein model:

$$H = \sum_i \frac{P_i^2}{2M} + \frac{1}{2}M\omega_0^2 X_i^2 - g\sum_i c_i^+ c_i X_i - t\sum_{\langle ij\rangle}\left(c_i^+ c_j + \text{H.c.}\right). \tag{1}$$

This model was originally devised to describe the interaction of charge carriers with local molecular vibrations, yet it can be considered as an effective model that captures qualitatively the interaction with other low-energy degrees of freedom such as the ones mentioned in the introduction. We study this model on a two-dimensional square lattice at half filling, representative of a generic low-dimensional

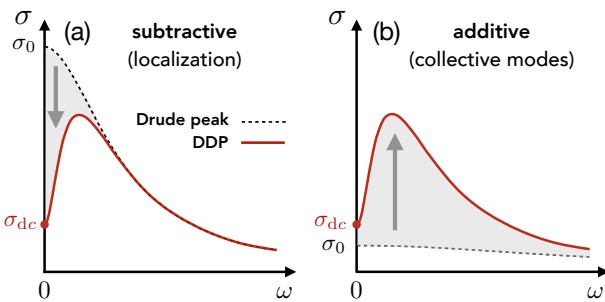

Figure 1: Two alternative microscopic scenarios leading to displaced Drude peaks. (a) Subtractive: a DDP emerges via a suppression of the metallic optical absorption at low frequencies; this can happen in the presence of slow bosonic scatterers, as considered in this work, or in the presence of static disorder. The d.c. conductivity $\sigma_{d.c.}$ is consequently reduced with respect to its semiclassical value $\sigma_0$. (b) Additive: the peak arises from an additional absorption channel, e.g. the direct absorption of light by collective modes, superimposed on the Drude response of electronic carriers.

metal, neglecting the spin degree of freedom which is irrelevant to our purposes. We consider the slow boson (adiabatic) regime in which the boson field can be safely considered to be classical. This approximation applies whenever the temperature $T \gtrsim \omega_0$ and $\omega_0/D \ll 1$; in this regime the system properties are governed by the dimensionless interaction parameter $\lambda = (g^2/2M\omega_0^2)/D$, $D = 4t$ being the half-bandwidth. Unless otherwise specified, we set $e = \hbar = k_B = 1$.

    We solve the model Eq. (1) employing two complementary theoretical approaches, whose comparison gives insight on the physical processes at play (more details in Appendix A): (i) Single-site dynamical mean-field theory (DMFT) in the adiabatic approximation that assumes static bosons, $\omega_0 \to 0$ ($M \to \infty$ with a fixed value of the spring constant $k = M\omega_0^2 = 1$). This provides a very accurate description of interaction effects regarding single-particle properties at all coupling strengths [21, 22]; it is however unable to describe localization, and the resulting transport mechanism is semi-classical, as appropriate in normal metals. (ii) Exact diagonalization on finite-size clusters (static-ED), also in the limit of static bosons. Crucial to the phenomenon we intend to address, while the bosons are treated classically owing to their slow dynamics (cf. Appendix A), here the electrons retain their full quantum-mechanical nature. This treatment therefore fully captures localization processes beyond the semi-classical regime. It provides an essentially exact determination of the optical absorption at all frequencies $\omega \gtrsim \omega_0$, because electrons responding faster than $\omega_0$ effectively see the boson field as a static, spatially-varying potential. Localization corrections are instead cut off at $\omega \lesssim \omega_0$ [14], with important implications as discussed below. Full details on the calculations, including the size of the clusters used in the exact diagonalization and the method used for thermalization of the bosons, are provided in Appendix A.

## 3  Results

**Disorder-driven displaced Drude peak (DDP).**    Fig. 2(a) shows the optical absorption of metallic carriers interacting with slow bosons, calculated from the model Eq. (1) in the weak/moderate coupling regime relevant to the majority of metals. The result from DMFT (brown, solid lines) shows textbook behavior, in the form of a conventional Drude-like absorption peak that broadens with in-

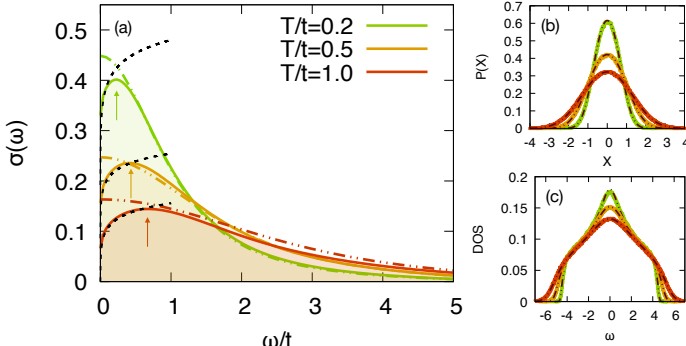

Figure 2: (a) Optical absorption of metallic carriers interacting with slow bosons in the regime of weak to moderate interactions, $\lambda = 0.3$. Solid shaded curves are static-ED results obtained upon averaging over 16000 static boson configurations on a $48x48$ lattice; the arrows mark the displaced Drude peak at $\omega_L$ (more details including finite size effects in Appendix A). Dash-dotted lines are DMFT results. Black dashed lines represent the weak localization result, Eq. ( [12]). The units of conductivity are given by $\bar{\sigma} = e^2/a\hbar$. Panels (b) and (c) show respectively the distribution of bosonic displacements $P(X)$ and the electronic density of states (DOS).

creasing temperature $T$: this is what is expected in the semi-classical picture, reflecting an increase of the scattering rate and the consequent increase of resistivity as the bosons are more and more thermally excited [23]. The static-ED result (colored, solid lines) agrees with the DMFT picture at high frequency, where it recovers the same Drude-like shape and the same temperature dependence. We note that static-ED includes all vertex corrections in the optical response, which are instead neglected within DMFT: the agreement shown in Fig. 2(a) therefore indicates that vertex corrections are negligible at high frequency within the model Eq. (1) (for contrasting results on the Hubbard model at strong interactions, see Ref. [8]). More importantly, the optical conductivity obtained from static-ED shows a downturn towards $\omega = 0$ not captured by DMFT, which results in a shift of the peak maximum to a finite frequency, $\omega_L$ (arrows). This behavior of the optical absorption, with the characteristic Drude absorption of quasiparticles being replaced by a displaced Drude peak, corresponds to the subtractive scenario sketched in Fig. 1(a).

To track the origin of the DDP, we present in 2(b) and (c) the distribution of local boson displacements, $P(X)$, and the electronic density of states (DOS). For moderate interactions the distribution $P(X)$ is Gaussian, representing the thermal fluctuations of the local site-potentials felt by the charge carriers. The resulting DOS for electrons moving in the fluctuating potential described by $P(X)$, shown in 2(c), is essentially a broadened version of the noninteracting DOS, with disorder-induced tails emerging at the band edges. Notably, no new features appear in the region near the chemical potential that is relevant for transport, here pinned at $\omega = 0$ due to particle-hole symmetry. The smooth and featureless nature of the electronic DOS implies that the dip observed at $\omega = 0$ in the optical conductivity does not originate from a modification of the single-particle properties. We note that for both the $P(X)$ and the DOS, the results from static-ED (solid) and DMFT (dash-dotted) perfectly coincide, further indicating that there are no relevant non-local effects in these quantities.

The above observations taken together are compatible with the DDP originating from electron localization in the random potential generated by the boson fluctuations: indeed, the quantum interference processes causing Anderson localization are entailed in the two-particle, current-current correla-

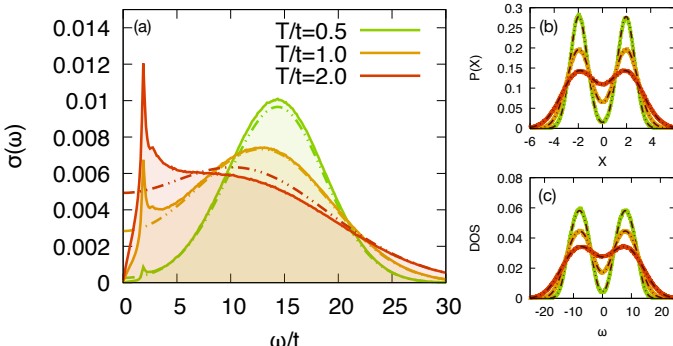

Figure 3: (a) Optical absorption of metallic carriers interacting with slow bosons in the regime of strong interactions, $\lambda = 2.0$. Solid shaded curves are static-ED results (16000 static boson configurations on a 48$x$48 lattice, cf. Appendix A). Dash-dotted curves are DMFT results. The units of conductivity are given by $\bar{\sigma} = e^2/a\hbar$. Panels (b) and (c) show respectively the distribution of bosonic displacements $P(X)$ and the electronic density of states (DOS).

tions, while leaving the DOS essentially featureless [14,24]. To prove this statement, we include in Fig. 2(a) the prediction of weak localization theory in two dimensions, $\sigma(\omega) = \sigma_0[1 + c \, (\hbar/t\tau) \log(|\omega\tau|)]$, with $\tau^{-1}$ the semi-classical scattering rate and $c$ a dimensionless factor ( [12–14] and Appendix C). Determining the scattering rate and the semi-classical conductivity $\sigma_0$ from the conventional high-frequency part of the spectrum leaves $c$ as the sole adjustable parameter. The weak localization result, shown as dashed lines in Fig. 2(a), is in perfect agreement with the calculated spectrum at low frequency (see also Fig. 8 in appendix C).

**Strong interactions: coexistence with the polaron peak.** It is now instructive to examine the optical absorption in the regime of strong electron-boson interactions, Fig. 3(a), which could occur for example in systems with extremely narrow bands or where the electron motion is already suppressed by other competing microscopic mechanisms. As in the weak coupling case, the optical absorption calculated via exact diagonalization shows a DDP in the low-frequency range, not captured by DMFT. Here, however, the weak localization formula used in Fig. 2(a) no longer describes the peak shape: the DDP at strong interactions takes the form of a narrow Lorentz oscillator peak emerging on top of the low-frequency normal absorption. The peak position is pinned at $\omega \simeq 2t$, which is indicative of the presence of strong bosonic disorder, as explained next.

The shortest localization length that can be achieved due to non-local interference effects corresponds to $L = a$ (wavefunction localized on two neighboring sites). In this limit an absorption peak will arise due to transitions between the bonding and anti-bonding states on such dimer, whose energy difference is $2t$ [25]. The observation of a peak at $\omega \simeq 2t$ in Fig. 3(a) is therefore indicative of the presence of strong bosonic disorder localizing the electronic wavefunction. The observed increase of spectral weight with temperature reflects the thermal activation of the initial (bonding) state, whose energy lies above the ground state.

In addition to the DDP, there is a second, broader peak at higher frequencies, signaling the formation of polarons — the electrons are bound together with the bosonic modes to form composite particles. This effect is embodied in a bimodal distribution of bosonic displacements [21], Fig. 3(b), which leads to the opening of a pseudogap already in the single-particle DOS, Fig. 3(c). Polaron formation, well captured by DMFT, results in a Gaussianly shaped optical absorption peak centered at

$\omega \approx 2E_P = 8\lambda t$, corresponding to the electronic transitions between the two maxima in the DOS [22] (similarly to what was shown in the weak interaction regime analyzed previously, vertex corrections to the optical conductivity are again irrelevant here at high frequency). Polarons melt thermally when $T \gtrsim E_P$, so that the polaron peak in $\sigma(\omega)$ progressively vanishes upon increasing the temperature, i.e. its trend as a function of temperature is opposite to that of the DDP.

**Peak position and localization length.** The temperature dependence of the peak frequency is illustrated in Fig. 4(a) for a wide range of interaction strengths (the values of $\lambda$ are indicated in the plot, filled/open symbols corresponding respectively to the weak and polaronic regimes illustrated in Figs. 2 and 3). At the lowest interaction strength analyzed here, $\lambda = 0.05$, the peak position follows a power law, $\omega_L \propto T^\alpha$ with exponent $\alpha \simeq 3/2$. The analysis of the weak localization correction provided in Appendix C indeed predicts that in this regime the peak position must scale with the scattering rate as $\omega_L \propto (\tau^{-1})^{3/2} \sqrt{|\log \tau^{-1}|}$, i.e. a power law with weak logarithmic corrections. The behavior observed in Fig. 4(a) follows from the fact that $\tau^{-1} \propto T$ for thermal bosons [21, 23].

As the interaction strength increases, the power law exponent is progressively reduced, until the curves become flat in the strong coupling regime (exponent $\alpha = 0$), corresponding to the pinning of the DDP to $\omega_L = 2t$ at strong interactions demonstrated in the preceding paragraph. Note that at large $\lambda$ the position of the DDP cannot be tracked down to the lowest temperatures because its weight becomes vanishingly small compared to the polaronic peak. The evolution with $\lambda$ shown in Fig. 4(a) implies that the DDP position does not follow a universal temperature dependence: all the exponents in the range $0 < \alpha < 1.5$ can in principle be encountered in materials (gray lines indicate the limiting slopes allowed by theory).

Interestingly, from the peak frequency we can obtain a direct estimate of the transient localization length $L$: as argued above it is equal to one lattice spacing $a$ in the strongly localized limit, where $\omega_L \simeq 2t$, and it obeys $L/a \simeq \sqrt{2t/\omega_L}$ when $\omega_L < 2t$. The transient localization length can therefore be accessed straightforwardly in an optical absorption experiment, using the known (calculated or measured) values of the transfer integrals $t$ for a given material. From the theoretical results shown in Fig. 4(a) we infer that $L$ does not exceed a few lattice spacings for all the explored interaction strengths and temperatures.

**Many-body enhancement of disorder.** The statistical variance of the local site-potentials, $s = g \sqrt{\langle X^2 \rangle}$, quantifies the amount of randomness associated with the bosonic fluctuations (cf. Eq. (1) and Figs. 2(b) and 3(b)). Having demonstrated that the DDP is caused by such randomness, it is expected that its position should be mostly governed by the variance $s$. This is indeed illustrated in Fig. 4(b), showing that the DDP frequency for all values of the interaction strength can be approximately scaled to a unique curve when plotted against this parameter. The success of the scaling hypothesis shows that the effect of disorder is almost entirely characterized by the local variance $s$, while the non-Gaussian nature of $P(X)$ at intermediate values of $\lambda$ and the existence of spatial disorder correlations can give rise to subleading corrections to scaling. Deviations from scaling at low values of $s/t$ are likely due to the incipient charge density wave order setting in at low temperature (see Fig. 5(a)).

The foregoing discussion implies that to rationalize the emergence and evolution of the disorder-induced DDP in bad metals it is crucial to understand the properties of the disorder variance $s$. For weak electron-boson interactions the feedback of the electrons on the boson fluctuations is negligible. Therefore, the amount of disorder increases with temperature $T$, because the fluctuations of the bosonic displacements follow directly from the equipartition principle, $\langle X^2 \rangle = T/(M\omega_0^2)$, leading to $s_0^2 = (8\lambda t)T$. As the interactions increase, however, the properties of the bosonic field are altered by

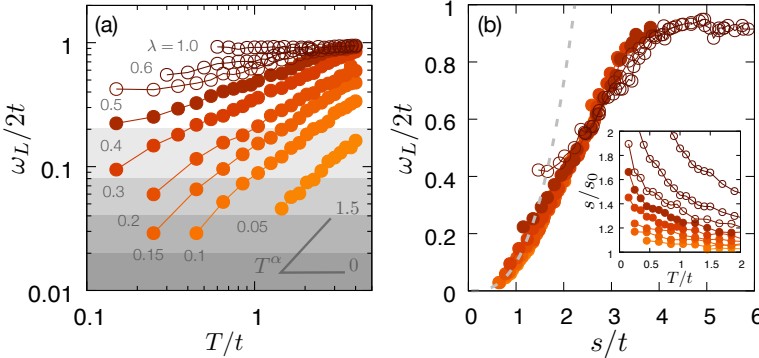

Figure 4: (a) DDP frequency, $\omega_L$, as a function of temperature (labels indicate the values of $\lambda$). The gray shaded areas correspond to different choices of the boson frequency $\omega_0/2t = 0.02, 0.04, 0.08$ and 0.2: the DDP disappears whenever $\omega_L < \omega_0$ (see text). (b) The same data plotted as a function of the local bosonic disorder $s$. The dashed line is the weak-localization estimate (see text). The inset shows the enhancement of disorder by many-body effects. In both panels open symbols refer to data in the strong coupling polaronic regime, $\lambda \geq 0.5$.

the presence of the electrons, whose self-consistent field causes growing anharmonicities in the potential of Eq. (1) [21]. This leads to a marked enhancement of disorder, that is especially strong at large $\lambda$/low $T$, as illustrated in the inset of Fig. 4(b). This many-body renormalization of the disorder potential has already been shown to enhance Anderson localization [19] and to cause resistivity anomalies in disordered metals [20]. As we show next, the same many-body mechanism is responsible here for a marked stabilization of the DDP at metallic densities when compared with the low-concentration limit (cf. Fig. 5).

As the anharmonicity of the potential grows, the distribution $P(X)$ deviates from Gaussian and eventually becomes bimodal in the polaronic phase at large $\lambda$, cf. Fig. 3(b). This implies that at large $\lambda$ a finite amount of randomness persists down to $T \rightarrow 0$, where the bosonic displacements act as a source of binary disorder (see also Appendix B). Such binary disorder is at the origin of the narrow DDP shown in Fig. 3(a).

As the preceding discussion suggests, the concept of a disorder-induced DDP is more wide-ranging than framework originally considered [5, 16], which relied on the existence of thermally fluctuating bosons: in fact, any source of slow randomness, even if its origin is non thermal, can also enable the phenomenon. Supporting this conclusion, a DDP persisting down to the lowest temperatures has been recently observed experimentally in proximity of the bandwidth-tuned Mott transition [10], possibly related to the presence of slowly fluctuating local moments (more examples in Fig. 6 below).

**Phase diagram of the many-body problem.**   The gray shaded areas in 5(a) show the boundaries separating the normal Drude metal from the anomalous DDP regime for different values of $\omega_0$. For weak to moderate interactions, a metal interacting with thermal bosons always features normal behavior at low $T$. Upon increasing the temperature, however, the boson-induced randomness unavoidably increases; if the interaction of the electron liquid with the boson fluctuations is significant, the system will acquire an anomalous behavior characterized by a displaced Drude peak. As shown in Fig. 5(a), the extent of the anomalous region depends on the dynamical scale $\omega_0$: while a finite frequency peak always exists in principle for static (extrinsic) disorder in two dimensions ($\omega_0 = 0$) [14, 24], when

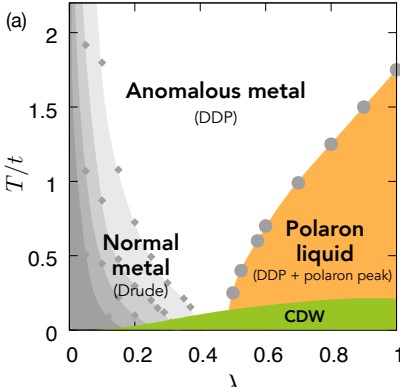
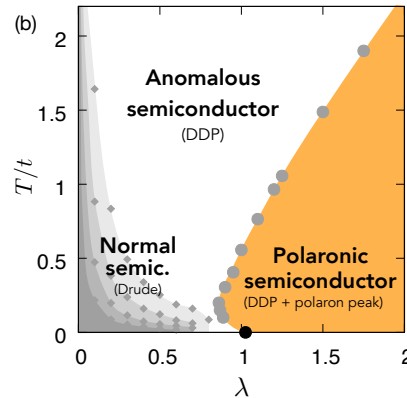

Figure 5: (a) Phase diagram summarizing the behavior of metals with slow dynamic disorder, corresponding to the solution of the model Eq. (1) at half filling. Gray shaded areas indicate the boundaries of the anomalous DDP region for different choices of the boson frequency, as in Fig. 4(a): from left to right (darker to lighter) $\omega_0/2t = 0.02, 0.04, 0.08, 0.2$. The full symbols are calculated by applying a Lorentzian broadening $p \simeq \omega_0$ to the optical conductivity obtained via the static-ED method, as described in Appendix A and benchmarked at length elsewhere [5, 26–28]). The polaron crossover (large dots) is signaled by the onset of a bimodal $P(X)$, cf. Fig. 3(b). The green area is the low-temperature CDW phase obtained in Ref. [29]. (b) Phase diagram of the same model in the limit of vanishing density, as appropriate to non-degenerate semiconductors. Here the polaron crossover is obtained from fully quantum DMFT [30] (see Appendix A).

the disorder is of dynamical origin the DDP is instead washed out as soon as $\omega_0 \gtrsim \omega_L$, causing the recovery of normal metallic behavior. The gray shaded areas in Fig. 5(a) were determined by applying a Lorentzian broadening of width $p = \omega_0$ to the static-ED optical conductivity and tracking the point where $d^2\sigma/d\omega^2|_{\omega=0}$ changes sign, as benchmarked in [28]; normal and anomalous behavior correspond respectively to $d^2\sigma/d\omega^2 < 0$ ($\sigma(\omega)$ peaked at $\omega = 0$) and $d^2\sigma/d\omega^2 > 0$ ($\sigma(\omega)$ peaked at $\omega > 0$).

At large values of the interaction strength, upon entering the polaron liquid phase at $\lambda > \lambda_P(T)$ (dots, orange shaded area), a peak of polaronic origin arises in addition to the DDP as shown in Fig. 3(a). The system eventually orders at low temperature into a CDW, whose study is however beyond the scope of this work (Fig. 5 reports the CDW transition obtained from DMFT in Ref. [29]).

Finally, we remind that the treatment used here applies to the classical boson regime where the temperature is larger than the bosonic energy: when quantum bosonic effects are restored, a Fermi liquid (metallic) state sets in at temperatures $T \lesssim \omega_0/4$ in the whole range $0 < \lambda < \lambda_P$, and whose optical absorption is characterized by a conventional Drude peak plus a bosonic side-peak at $\omega \simeq \omega_0$ [30, 31].

**Comparison with the low-concentration limit.** The DDP problem in the low-concentration limit appropriate for non-degenerate semiconductors can be solved using the same static-ED method that we have used at metallic densities: here we consider a single electron in the diagonalization procedure instead of a finite concentration of electrons. The resulting phase diagram is shown in Fig. 5(b). Comparison with Fig. 5(a) shows that the overall features of the phase diagram are retained regardless of the electron density (except for the CDW phase, which cannot develop at vanishing den-

sities). In particular, as already reported elsewhere [5, 26–28, 31], also in the low concentration limit there exists a pervasive DDP regime, whose origin is therefore common to metals and non-degenerate semiconductors. An important difference revealed by Fig. 5, however, is that at metallic densities the DDP region is significantly more stable than at vanishing densities in terms of interaction range (gray symbols/lines in Fig. 5(a) and (b)). A similar stabilization occurs also for the polaronic phase.

The critical value $\lambda_P(0) \simeq 0.5$ for the polaron crossover at half filling is approximately half that obtained for an individual electron, $\lambda_P(0) \simeq 1.0$. This reflects the fact that many-body effects favor self-trapping due to an appreciable back-reaction of the charge density on the bosonic fluctuations [21]. The self-consistent (Hartree) field responsible for this enhancement is instead absent at vanishing electronic densities.

Comparison of Figs. 5(a) and (b) shows that this many-body mechanism, enabled at sufficiently strong interactions, also enhances the DDP phase, via the renormalization of the disorder potential discussed in the previous paragraphs (inset of Fig. 4(b)). By effectively increasing the amount of randomness, the back-reaction of the electron density on the fluctuating potential favors localization, hence promoting the emergence of the DDP to lower values of the interaction strength.

## 4 Discussion and conclusions

The theoretical prediction of a DDP for a wide range of microscopic parameters, including moderate values of the electron-boson interaction $\lambda$ as can be expected in metals, is compatible with the widespread experimental observation of DDPs in various classes of materials. To illustrate this, Fig. 6 reports the temperature dependence of the DDP frequency $\omega_L$ measured in different classes of compounds, including the cuprate superconductors, various transition-metal oxides, layered organic conductors and Kagome metals [6, 10, 32–36]. While a large scatter is observed in the absolute values, which is expected due to the very different energy scales characterizing these compounds (whose bandwidths range from fractions of eV to several eV), the peak position is invariably an increasing function of $T$ at high temperature, demonstrating a prominent role of thermal fluctuations as expected within the present framework. The chosen logarithmic scale reveals a large variability of exponents $\omega_L \propto T^\alpha$, in agreement with the theory; all the experimentally observed exponents fall in the range predicted in Fig. 4(a), i.e. $0 < \alpha < 3/2$ (gray lines).

The saturation of the peak position observed at low temperature in some compounds is in principle compatible with the strong coupling behavior shown in Fig. 4(a) (open circles), but this explanation is not supported by the concomitant observation of a polaronic peak in the measured optical spectra. An alternative explanation for the saturation could then be the presence of extrinsic (e.g. $T$-independent) sources of disorder, which would dominate at low $T$ when the thermal disorder becomes negligible. Another possibility is that what is actually reported at low $T$ is a bosonic side-peak located at $\omega \simeq \omega_0$, signaling the recovery of the conventional weak-coupling picture in the quantum limit $T \lesssim \omega_0/4$ [30, 31].

As our results have demonstrated, a sufficient requirement for the emergence of a localization-induced DDP is the presence of slowly fluctuating degrees of freedom interacting with the electronic carriers. This condition is likely to be met in a variety of physical situations, as microscopic candidates are ubiquitous in complex materials — including soft lattice vibrations [5], critical modes in proximity of phase transitions [11], fluctuating magnetic moments [37], or collective charge fluctuations [33, 38, 39]. In this respect, charge fluctuations induced by the long-range Coulomb interactions between electrons may provide the slow bosonic modes enabling the appearance of a DDP [10, 38, 39], while

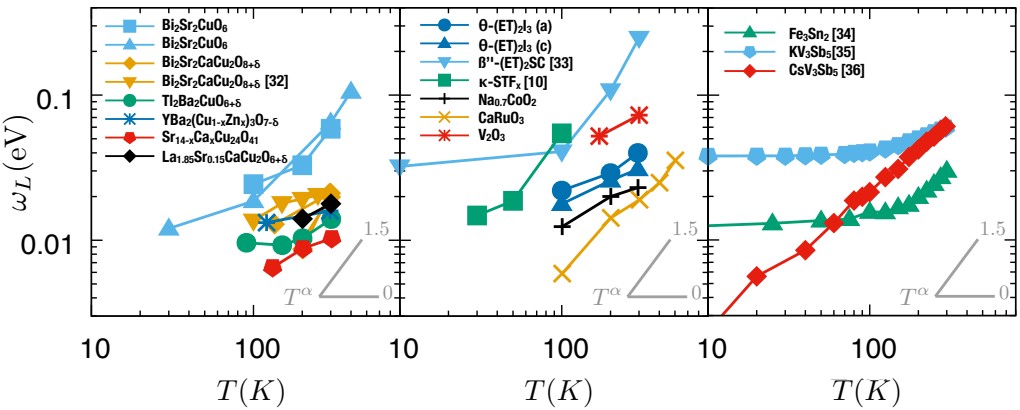

Figure 6: Temperature dependence of the DDP frequency in various materials (left panel: cuprate superconductors; central panel: organic conductors and other oxides; right panel: kagome metals). Unless otherwise indicated (Refs. [10, 32–36]), the datasets are the ones collected in Ref. [6].

short range correlations alone do not seem to sustain a DDP feature in the optical absorption [7–9]. On general grounds, local repulsive interactions are expected to suppress charge fluctuations and other forms of self-generated randomness, in particular at integer fillings. How these effects compete both in parent and doped strongly correlated systems remains as an open question.

Although a full quantitative exploration of the temperature dependent resistivity within the DDP regime is beyond the scope of this work, we briefly speculate here on the broader consequences of the present findings on charge transport. Because the conductivity in the d.c. limit and the low-frequency optical absorption are intimately connected, the quantum localization corrections at the origin of the DDP phenomenon demonstrated here are also expected to strongly alter the charge transport mechanism, as is now well established in the field of organic semiconductors [5, 17]. As inferred from the optical conductivity, in the presence of dynamic disorder, the electronic wavefunctions show a marked tendency to localization at short times $t \lesssim 1/\omega_0$, leading to a sizeable suppression of the charge conductivity, cf. Figs. 1(a) and 2(a). The interaction with slowly fluctuating degrees of freedom therefore provides a general route to bad metallic behavior, along with more commonly explored scenarios such as strong correlations and quantum criticality [7, 9, 40, 41]. It will be interesting to analyze theoretically the DDP formation arising in other microscopic models, where the existence of slow bosons is not assumed from the start as we have done here, but rather emerges from the many-body electronic interactions.

# Acknowledgements

The authors thank M. Binet, L. de' Medici, V. Dobrosavljević, M. Dressel, B. Goutéraux, A. Pustogow and E. Uykur for stimulating discussions and valuable input.

# A   Calculation details

We solve the model Eq. (1) employing two complementary theoretical frameworks, whose comparison is extremely informative on the physical processes at work:

(i) Single-site dynamical mean-field theory (DMFT), which provides a very accurate description of interaction effects contained in the single-particle properties, at all coupling strengths [21, 42]. Since this version of DMFT lacks the non-local interferences encoding localization processes in the two-body response function (vertex corrections), the resulting charge transport is in essence semi-classical. To address the regime of slow bosons we take the adiabatic approximation which assumes classical bosons and results in a self-consistent version of the coherent potential approximation, correctly describing the polaronic state [21].

(ii) Exact diagonalization of the model Eq. (1) on finite-size clusters, also in the limit of static bosons (static-ED). This treatment too fully describes interaction effects in the one-particle properties, including polaron formation. In addition, it is able to capture the localization of the wavefunction originating from non-local interference processes beyond the semi-classical limit, contained in the two-body current-current correlation function. In practice we implement a Langevin approach for the bosonic displacement supplemented by an acceptance check [43], [44]. The Langevin equation for the oscillator's coordinate is evaluated via the Euler's approximant

$$X_i' = X_i + f_i(X)\Delta t + \sqrt{2T\Delta t}g_i \tag{2}$$

where $T$ is the temperature, $\Delta t$ the time step and $g_i$ is a normal Gaussian number, and we use units such that $M = k = 1$. In Eq. (2) the force $f_i(X)$ (we use the shortcut notation $X = \{X_i\}$ for the full coordinate set) is given in terms of the electron densities $\langle n_i \rangle$ as

$$f_i(X) = -X_i - g(\langle n_i \rangle - n). \tag{3}$$

In Eq. (3) the local electronic density is obtained via exact diagonalization, using standard linear algebra routines (`LAPACK`), of the electronic system at a given configuration of displacements and $n$ is the average electron density at equilibrium which is $1/2$ for spinless electrons at half filling. The proposed configuration $X'$ is accepted if a uniform number in $[0, 1]$ is less than the probability ratio

$$\frac{P(X|X')P(X')}{P(X'|X)P(X)} = \exp(-\beta\Delta W(X, X')) \tag{4}$$

where $X$ is the actual coordinate set and $X'$ is the proposed update according to Eq. (2). The function $W(X, X')$ is given by

$$\begin{aligned} W(X, X') &= F(X') - F(X) + \\ &+ \frac{1}{2}\sum_i (X_i' - X_i)(f_i(X') + f_i(X)) + \\ &+ \frac{1}{4}\Delta t \sum_i (f_i^2(X') - f_i^2(X)) \end{aligned} \tag{5}$$

and $F(X)$ is the thermodynamic potential of the electronic system at a given configuration of $X$. In this scheme the time-step $\Delta t$ is an adjustable parameter that can be chosen in order to optimize the thermalization process.

The data shown in Fig.2 of the manuscript and in Fig. 7 and 9 are obtained by thermalizing the system for a Langevin time $\tau_{therm} = 100$ in units of the oscillator frequency $\omega = \sqrt{k/M}$. The optical

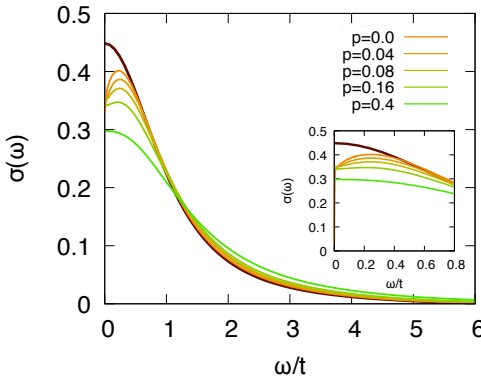

Figure 7: Optical conductivity for $\lambda = 0.3$, $T = 0.2t$. DMFT (brown), static-ED and RTA-ED at different values of the relaxation parameter $p$ (color, labels in units of $t$). The static-ED is recovered for $p = 0$, with $\sigma(0) = 0$. The inset shows the low frequency part of the spectrum.

.

conductivity has been calculated using the Kubo formula and averaging over up to 16000 oscillator configurations. The static-ED method is numerically affordable and it provides a very accurate determination of the optical absorption properties at frequencies $\omega \gtrsim \omega_0$: electrons oscillating faster than the frequency scale of the bosons effectively see the latter as a static spatially-varying potential, so that in this range the spectrum calculated assuming $\omega_0 = 0$ is essentially exact. The d.c. conductivity instead cannot be addressed by the static-ED method, since electron localization by a static random potential in dimensions $d \leq 2$ implies $\sigma(0) = 0$.

In order to restore the boson dynamics that become important at $\omega \lesssim \omega_0$ we supplement the static-ED method by a relaxation time approximation (RTA-ED), as described and benchmarked at length elsewhere [5, 26, 28]. The RTA-ED in its simplest version amounts to applying a Lorentzian broadening of width $p \simeq \omega_0$ to the static-ED result. [5, 27], as illustrated in Fig. 7.

A fully quantum DMFT treatment of the problem has been employed to obtain the numerical data for the polaron crossover presented in Fig. 5b). In this case we use the codes developed for the optical conductivity of a single-polaron [30] adapted to the square lattice. We use a finite value of $\omega_0/t = 0.4$ which is well inside the adiabatic regime. As the distribution of a generic displacement is not affected by the interaction with a for a single electron we find the crossover by looking at the development of finite frequency peak in the optical conductivity as a function of the temperature and the coupling constant $\lambda$. A sample of optical conductivity in the adiabatic regime can be found in [30] fig. 3b in the Bethe lattice case. To overcome the difficuties in locating precisely the polaron peak in the optical conductivity due to the presence of multiphonon peaks at least at low temperature we use a Reik's formula with adjustable parameters (Eq. (12) ref. [30] ). A sample of typical optical conductivity obtained in the regime near the polaron crossover is shown toogether with Reik's formula fitting in fig. 8 The $T = 0$ polaron crossover (black dot in fig.5b)) has been obtained by inspecting the square-lattice ground state energy obtained trough DMFT using the same value for the adiabatic parameter $\omega_0/t = 0.4$. A sample of typical beahviour for this quantity can be found in [45] fig. 4.

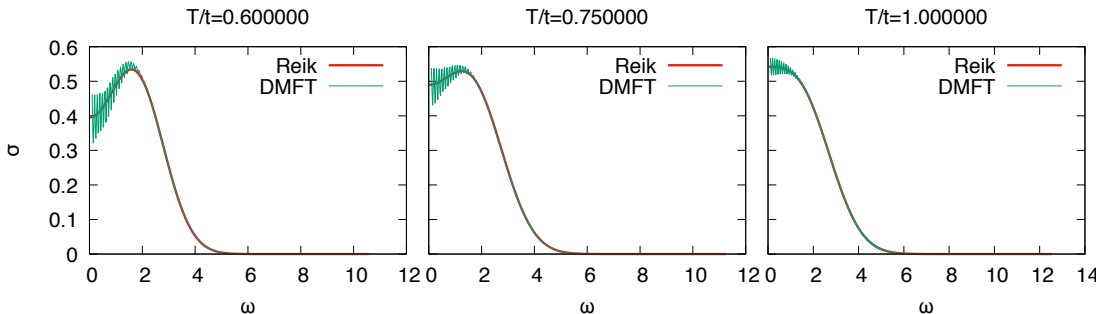

Figure 8: Optical conductivity for a single polaron at $\lambda = 1.2$ and $\omega_0/t = 0.4$ at different temperatures .

## B  Solution of the two site model

The Holstein model for two sites ($k = M\omega_0^2$):

$$
\begin{aligned}
H &= \frac{P_1^2 + P_2^2}{2M} + \frac{1}{2}k(X_1^2 + X_2^2) - gc_1^+c_1X_1 + c_1^+c_2X_2 \\
&\quad - t\left(c_1^+c_2 + \text{H.c.}\right),
\end{aligned}
\tag{6}
$$

can be rewritten in the form

$$
H = \frac{P_G^2}{4M} + \frac{P^2}{2M} + \frac{1}{2}kX_G^2 + \frac{1}{2}kX^2 - gX_G - g\sigma_z X - t\sigma_x
\tag{7}
$$

by introducing the boson center of mass $X_G = (X_1 + X_2)/2$ and relative coordinate $X = (X_1 - X_2)/2$ and considering one spinless electron on two sites. In Eq. (7) the Pauli matrices represent the pseudo-spin $\sigma_x = c_1^+c_2 + c_2^+c_1$ and $\sigma_z = c_1^+c_1 - c_2^+c_2$. It is worth to note that despite the small system size, one spinless electron on two sites represents a half filled system. Omitting the trivial center-of-mass part of $H$ the eigenvalues in the adiabatic limit for the relative boson $X$ are

$$
E_{\pm}(X) = \frac{1}{2}kX^2 \pm \sqrt{(gX)^2 + t^2}.
\tag{8}
$$

This yields the following thermal distribution for the relative variable

$$
P_r(X) = \mathcal{N}\exp(-\beta kX^2)\cosh(\sqrt{(gX)^2 + t^2}),
\tag{9}
$$

with $\mathcal{N}$ being a normalization constant. Defining the site energy as $\epsilon_i = gX_i = g(X_G \pm X)$ we can write its local fluctuation as the sum $s^2 = s_G^2 + s_r^2$ with

$$
s_G^2 = g^2\langle X_G^2\rangle
\tag{10}
$$

$$
s_r^2 = g^2\langle X^2\rangle
\tag{11}
$$

being $X_G$ and $X$ independent variables. In terms of the dimensionless coupling $\lambda = g^2/2kt$, it is straightforward to calculate $s_r^2 = s_0^2 = 2\lambda tT$ for $\lambda \to 0$. For finite but weak interaction strengths, the Gaussian appearing in Eq. (9) is modified by the cosh term giving

$$
P_r(X) = \mathcal{N}\exp(-\beta k_{eff}X^2)
\tag{12}
$$

with $k_{eff} = k[1 - \lambda \tanh(\beta t)]$. As a consequence $s_r^2 = T/k_{eff}$

$$s_r^2 = \frac{s_0^2}{1 - 2\lambda \tanh(\beta t)}. \tag{13}$$

The previous expression diverges at $\lambda = \lambda_P(T)$, where

$$\lambda_P(T) = \frac{1}{2 \tanh(\beta t)} \tag{14}$$

and $\lambda_P(0) = 1/2$ at zero temperature. This divergence is a signal of the bimodal nature of $P_r(X)$, which marks the polaron crossover. The prediction Eq. (14) agrees remarkably well with the ED solution on extended clusters reported in Fig. 5.

For strong coupling or large temperatures $T \gg t$ we can take the atomic limit ($t = 0$) to estimate $s_r^2$. In this limit the distribution of $X$ becomes

$$P_r(X) = \mathcal{N} \exp(-\beta k X^2) \cosh(|gX|). \tag{15}$$

This is the sum of two Gaussians of variance $T/k$ centered in $X = \pm X_0$ with $X_0 = g/k$. Evaluation of the fluctuations yields

$$s_r^2 = s_0^2(1 + 2\lambda\beta t). \tag{16}$$

Note that as $T \to 0$ the previous formula gives a finite $s_r^2 = 4\lambda^2 t^2$. A finite $s_r^2(T = 0)$ persists beyond the atomic limit as long as $\lambda > 1/2$: a direct calculation gives

$$s_r^2(T = 0) = t^2(4\lambda^2 - 1) \tag{17}$$

For $\lambda < \lambda_P(T = 0)$, $s_r^2$ smoothly interpolates from Eq. (13) to Eq. (16) while for $\lambda > \lambda_P(T = 0)$ a finite residual value of local energy fluctuations Eq. (17) persists at $T = 0$. To obtain the total site energy fluctuations one has to add the interaction-independent center-of-mass contribution $X_G$, i.e. $s_G^2 = 2\lambda T$.

## C  Peak position within the scaling theory of localization

An estimate of the peak position using the scaling theory of localization can be obtained by equating the optical conductivity in the scaling regime [12, 14]

$$\sigma_{2D}(\omega) = \sigma_0(1 + \frac{1}{k_F \ell} \log(|\omega\tau|)) \tag{18}$$

with the diffusive Boltzmann form that is valid at sufficiently large frequencies in the normal metal:

$$\sigma_{SC}(\omega) = \frac{\sigma_0}{1 + (\omega\tau)^2}. \tag{19}$$

In Eqs. (18,19) $k_F$ is the Fermi momentum, $\ell$ is the mean free path and $\tau$ is the semi-classical scattering time. A comparison of these two approximations and our numerical data is shown in Fig. 9 for weak coupling and low temperature, as well as in Fig.2(a). In the right panel of Fig. 9 a finite size effect is revealed by comparison of the numerical data obtained using different lattice sizes. Finite size effects are absent in the peak region and above.

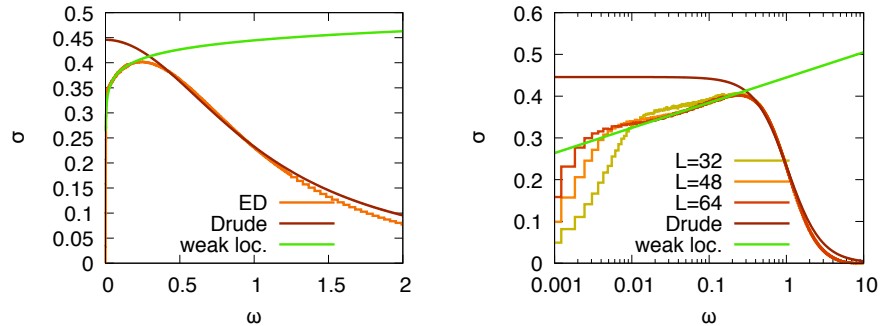

Figure 9: Left panel: comparison of the analytical results Eq. (18) (weak localization) and Eq. (19) (Drude) with the ED result for $\lambda = 0.3$ and $T = 0.2$. The parameters $\tau$ and $\sigma_0$ were fitted from exact diagonalization on a 48x48 square lattice at sufficiently large frequencies (here $0.3 < \omega/t < 2.0$). The parameter $\xi$ left as the only free parameter in Eq. (19) was fitted from the low energy behaviour of the ED spectrum at frequencies below the peak (here $0.03 < \omega/t < 0.13$). Right panel: illustration of finite size effects in the low energy behaviour of the optical conductivity on 32x32,48x48 and 64x64 square lattices. In both figures raw data are plotted as histograms as they were calculated. Continuous lines shown in previous figures and in the main text are obtained by joining the center of each histogram bin.

.

Letting $x = \omega\tau$, we solve the following transcendental equation with $\xi = k_F\ell$ as a parameter

$$x = \exp(-\frac{\xi x^2}{1 + x^2}). \tag{20}$$

The solution is obtained by expressing $\xi$ as a function of $x$,

$$\xi = -\frac{1 + x^2}{x^2} \log(x). \tag{21}$$

The function $x(\xi)$ is plotted in Fig. 10.

As $\xi \gg 1$ we can expand the r.h.s. to lowest order in $x$ and subsequently evaluate the log term, which yields

$$x \simeq \frac{\log^{1/2}(\xi)}{\sqrt{2\xi}} \quad \xi \gg 1. \tag{22}$$

In the opposite limit $\xi \ll 1$, $x \to 1$ therefore

$$x = 1 - \frac{\xi}{2} \quad \xi \ll 1. \tag{23}$$

By taking into account the next order in the $\xi \gg 1$ expansion we obtain

$$x \simeq \left[\frac{2\xi}{\log(1 + \xi)} - 1\right]^{-1/2}. \tag{24}$$

This expression interpolates between $\xi \gg 1$ and $\xi \ll 1$ as shown in Fig. 10, providing an analytical formula for the peak position as a function of the scattering time.

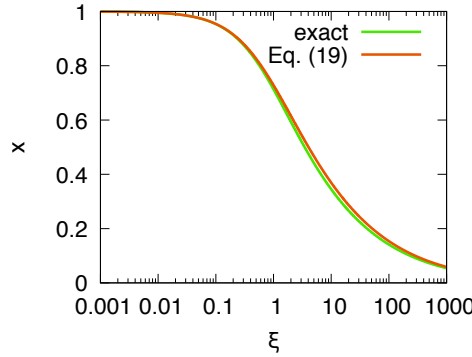

Figure 10: The function $x(\xi)$ and the approximation Eq. (24)

.

The knowledge of the function $x(\xi)$ allows to estimate the temperature dependence of the peak position $\omega_L$. To be consistent with the range of applicability of the present formulas we consider the weak disorder case, $\xi \gg 1$. From perturbation theory $\tau, \ell \propto t/s^2 \propto 1/\lambda T$, where $s$ is the local energy fluctuation (see main text and Appendix B). In the limit of large mean-free-path we therefore have $\omega_L \propto (\lambda T)^{3/2} \log^{1/2}(1/\lambda T)$, i.e. a 1.5 temperature exponent plus logarithmic corrections.

The analytical form for the peak position as a function of $\xi$ Eq. (1) also implies a scaling behavior for the curves of peak position against the temperature. This is obtained by setting

$$\xi = \frac{\Delta^2}{s^2} \tag{25}$$

where $\Delta \propto t$ sets the energy scale of the model. Using Eq. (25) it is possible to estimate $\omega_L$ as

$$\omega_L = A\frac{\Delta}{\xi}\left[\frac{2\xi}{\log(1+\xi)} - 1\right]^{-1/2} \tag{26}$$

with $A$ a numerical constant. Analytical estimates of the peak position can be obtained by assuming $\Delta = 2t$, $A = 1$ in the equation above.

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
