# Peer review of "Displaced Drude peak and bad metal from the interaction with slow fluctuations."

_SciPost Physics_

## Round 2 · Referee Report · Anonymous · 2021-6-22

Strengths
1. Solid miscroscopic description of a generic macroscopic observable behavior.
2. Solidity of the numerical and analytical insights to the problem.
3. Generality of the results with respect to different regimes of parameters and applicability to real materials.
Weaknesses
1. Ad-hoc modelling of the bosonic degrees of freedom. Discussed in the perspectives.
2. Lack a discussion of the possible electronic interaction effects.
Report
In this manuscript the authors investigate the effects of the coupling to slow bosonic fluctuations onto the optical conductivity of an electronic system. In particular, the authors demonstrate that quantum localization corrections associated to such coupling are a sufficient condition to realize an energy displacement of the Drude peak (DDP), as observed experimentally in different compounds. Remarkably, not only it is pointed out that such DDP effect is particularly stable for metals but the authors also provide a detailed comparison of the results to a large number of materials of different origin (Cuprates, organic conductors, Co or Ru Oxides, etc.).
The subject of this work is very interesting and timely, proposing a novel explanation to a well known issue.
The authors make use of different and somehow complementary approaches based on the exact diagonalization algorithm within a static approximation, which is valid in the adiabatic regime. The authors discuss the advantages and the limitations of the two methods in different contexts.
The manuscript is very well organized and clearly written. The analysis at weak- and strong-coupling is complemented with analytic insights (extensively discussed in the appendices), while the scaling properties at intermediate coupling are rationalized in terms of their dependence with respect to disorder and/or coupling.
The results presented in this work are solid and their discussion is detailed and precise.
This work definitively meets all the acceptance criteria of the journal, as such I feel to recommend its publication. Yet, I do have some minor questions, doubts or comments, which I list below:
* The authors seem to never explicitly discuss some of the details of the numerical calculations, like the number of sites/levels used in the exact diagonalization algorithm, whether they are relying on the full spectrum or just its extremal part (Lanczos), etc. This is useful to establish the accuracy of the calculation with respect to the discretization or possible finite size effects, (which I assume are mitigated by the statistical average).
* The basic modelling used in this work is essentially a Holstein system. One can reasonably expect the presence of electronic interaction to partially modify the results, either quantitatively or even qualitatively. Can the authors briefly comment on this aspect? This seems to be particularly relevant to make more adherent comparison with materials which are known to be strongly correlated, e.g. Cuprates.
* In fig.2 and fig.3 can the author report the specific values of \omega_0 (and g) determining the adimensional \lambda term?
* Concerning fig.4:
i) What do indicate open and filled symbols in panel a)
ii) as a rule of thumb the scaling of some quantity usually requires it to span one or even two order of magnitude variation fo the independent variable (here the Temperature). While most of the data at intermediate values of \lambda do fulfil this "rule", other results for smaller or larger \lambda are much more limited in their variation. Why?
* At pag.5, third line the authors discuss the strong coupling regime and write "Because of bosonic disorder is now strong...". However, this sentence (although correct) somehow anticipates the forthcoming results concerning P(x) and
s/s_0 and the many-body enhancement of the disorder.
* At pag.6, last paragraph the authors discuss the limiting case g=0 (absence of electron-boson coupling). I guess the resulting expression for s_0 is valid for \lambda-->0 not strictly for g=0.

---

## Round 2 · Referee Report · Anonymous · 2021-6-28

Strengths
1) Expansion of the transient localization scenario to cover bad metals
2) Interesting analytical and numerical discussion
3) Comparison with results for real materials
Weaknesses
1) The algorithm used should be better described
2) Some acronyms and symbols should be better defined
Report
In the paper “Displaced Drude peak and bad metal from the interaction with slow fluctuations” the authors apply the well-known transient localization scenario, which had high success in the field of organic semiconductors, to the case of bad metals to explain the appearance of a displaced Drude peak (DDP). Using two different models (dynamical mean field theory or exact diagonalization of clusters), the authors highlight that the DDP is recovered only going beyond the semi-classical regime taking into account localization effects. Notably, the DDP is used to evaluate the transient localization length and a comparison of results for several different materials is provided.
In general, the manuscript is interesting and clear, providing both numerical and analytic insights in the appendices. My opinion is that the paper can be accepted after minor revisions listed below. Further revision is not needed.
Requested changes
1. In pages 2-3, I cannot find what is “T”. I think (from what I found in the appendices) that T is the temperature and that maybe in figures 2-3 the adimensional T values reported in the panels are in terms of “t” (the coupling), i.e. T=0.2t and so on, but I think you should discuss this point in the main text (not only the appendices).
2. Caption of fig. 2. Even though it is said in the text, I think you should say also in the caption that the dashed lines represent the weak localization.
3. Figure 4. Why some points are filled circles, while other are open circles? What is the difference?
4. The acronym CDW should be defined.
5. Maybe I am missing something, but I cannot find details of the numerical calculations (at least for figures in the main text), e.g. the size of the finite cluster (which is on the contrary provided for fig 8 in the appendices) or the algorithm used for the diagonalization part.
6. Just a personal opinion: the figures appear a bit difficult to read as the colours are not well chosen. For example
• in fig. 2 and 3 the brown solid line (DMFT results) for all three curves is a bit confusing, you could for example replace it with dotted lines of the color of each ED curve.
• in fig. 3 it is quite difficult to visualize the yellow peak under the broad red shaded area.
7. There are some typos in the references, please check them. For example, in ref. 30 one author is “S.F.” (I assume it stands for Simone Fratini)

---

## Editorial Decision

unknown